# Prevalence of Worry-Induced Sleep Disturbance and Associated Factors among a National Sample of In-School Adolescents in Lebanon

**DOI:** 10.3390/bs10100148

**Published:** 2020-09-27

**Authors:** Supa Pengpid, Karl Peltzer

**Affiliations:** 1ASEAN Institute for Health Development, Mahidol University, Salaya, Phutthamonthon, Nakhon Pathom 73170, Thailand; supa.pen@mahidol.ac.th; 2Department of Research Administration and Development, University of Limpopo, Sovenga 0727, South Africa; 3Department of Psychology, University of the Free State, Bloemfontein 9300, South Africa

**Keywords:** worry, sleep disturbance, risk behaviour, protective factors, adolescents, Lebanon

## Abstract

Persistent worry can cause significant distress among adolescents. The goal of this study was to estimate the prevalence and correlates of worry-induced sleep disturbance (WISD) among adolescent school children in Lebanon. Cross-sectional, nationally representative data were analysed from 5849 adolescents (15 years median age) that took part in the “2017 Lebanon Global School-Based Student Health Survey (GSHS)”. The results indicate that the prevalence of WISD was 14.7%, 9.6% among males and 17.2% among females. In adjusted logistic regression analysis, loneliness, older age, female sex, having no close friends, infrequent bullying victimization, parents disrespected privacy, current tobacco use, ever cannabis use, high leisure-time sedentary behaviour and having sustained multiple serious injuries (past year) were associated with WISD. In addition, in unadjusted analysis, mostly or always feeling hungry (or low economic status), school truancy, having been physically attacked, frequently being in physical fights (past year), low peer support, parental emotional neglect, parents never checking homework, ever drunk and frequent soft drink intake were positively, and infrequent fast food intake was negatively, associated with WISD. One in seven students reported WISD and several associated factors were identified, which can aid intervention strategies. Multi-level interventions are needed targeting psychosocial distress, social-environmental factors and health risk behaviours to prevent WISD in this adolescent school population.

## 1. Background

“Mental health conditions, including depression and anxiety, account for 16% of the global burden of disease and injury in people aged 10–19 years.” [1]. In children and adolescents, the “worldwide-pooled prevalence of mental disorders was 13.4%, including any anxiety disorder 6.5% and any depressive disorder 2.6%” [2]. “Generalised anxiety disorder in a child or adolescent is excessive worry and tension about everyday events that the child or adolescent cannot control and that is expressed on most days for at least 6 months, to the extent that there is distress or difficulty in performing day-to-day tasks.” [3] Poor and inadequate sleep constitute an important health problem among adolescents, having negative effects on mental and physical health [4]. Due to psychosocial and physiological reasons, generally satisfactory sleep changes from prepuberty to less satisfactory sleep during adolescence [5]. Reasons for the latter include worries, depression, anxiety, substance use, caffeine-containing drinks, and others [5]. “During adolescence, worry becomes more prominent with the development of abstract thinking and the cognitive ability to foresee multiple negative outcomes” [6]. Worry is a cognitive component of anxiety [6]. Worry-induced sleep disturbance (WISD) is defined as mostly or always “worried about something that you could not sleep at night in the past 12 months” [7]. Among children and adolescents, in moderation, worry may be adaptive, but if worries persist over time, they can cause clinical concern, and pathological worry has been associated with various problems, including poor academic performance and disrupted sleep [8].

In a previous multi-country adolescent school survey [7], the overall prevalence of WISD was 7.8%, including in Kuwait 18.9% and Qatar 17.9%. In the 2011 Lebanon “Global School-Based Student Health Survey (GSHS)” the prevalence of WISD was 10.3% [8]. In a local survey among secondary school students in Beirut, Lebanon, a large proportion of students had poor sleep quality (76.5%) [9], and in household survey among adolescents (N = 510) in Beirut, 26.1% had 30-day psychiatric disorders, including 13.1% anxiety disorders [10]. There is a lack of more recent national data on the prevalence and correlates of WISD among adolescents in Lebanon.

Factors associated with WISD among adolescents can be divided into psychosocial distresses, socio-environmental factors and health risk behaviours [11]. Psychosocial distresses associated with WISD or psychological distress may include loneliness [12], having no close friends [13], interpersonal violence [13], and bullying victimization [12]. Socio-environmental factors associated with WISD or psychological distress, may include experience of hunger [14], low peer and low parental support [13,14,15,16] and school truancy [15]. Health risk behaviours associated with WISD or psychological distress, may include substance use [14,17,18], sedentary behaviour [8,19], frequent fast food consumption [20], frequent soft drink intake [20], fast food intake and sedentary behaviour [21], sexual behaviour [14] and injury [19]. This investigation aimed at estimating the prevalence and correlates of WISD among adolescents in Lebanon.

## 2. Methods

### 2.1. Sources of Data

Cross-sectional nationally representative data from the “2017 Lebanon GSHS” were analyzed [22]. More detailed information about the survey methods and the data can be sourced [22]; the overall response rate was 82% [22].

### 2.2. Measures

The questionnaire used is shown in Appendix A. Worry-induced sleep disturbance was assessed with the question, “During the past 12 months, how often have you been so worried about something that you could not sleep at night?” Response options were never, rarely, sometimes, most of the time or always. Most or the time or always was defined as worry-induced sleep disturbance, while never, rarely or sometimes was defined as no worry-induced sleep disturbance, in line with previous research [7]. Emotional neglect was defined as never “parental or guardian understanding of your problems and worries? And parents or guardians never really know what you were doing with your free time when you were not at school or work?” [23].

## 3. Data Analysis

Statistical analyses were done with “STATA software version 15.0 (Stata Corporation, College Station, TX, USA).” Unadjusted and adjusted (with variables significant in unadjusted analysis) logistic regression analyses were used to assess predictors of WISD. Missing values were not included in the analysis. *p* < 0.05 was accepted as significant.

## 4. Results

### 4.1. Sample and Worry-Induced Sleep Disturbance Characteristics

The sample comprised 5708 school adolescents (15 years median age, 3 years interquartile range), and 56.9% were female. In terms of psychosocial distress, 13.1% of adolescents were lonely, 4.6% had no close friends, 5.3% had frequently been bullied (≥3 days/past month), 10.7% had frequently been attacked (≥2 times/past year) and 20.1% had frequently been in a physical fight (≥2 times/past year). In relation to social-environmental factors, 2.6% of students were mostly or always hungry, 36.8% were exposed to daily secondary smoke, 17.3% were school truant, 20.0% had low peer support, 11.1% experienced parental emotional neglect, 23.9% had parents who never checked on their homework, and 7.6% had parents who mostly or always disrespected their privacy. Regarding health risk behaviours, 35.9% of the students used currently tobacco, 10.9% had ever been drunk, 2.5% had ever used cannabis, 13.0% had three or more soft drinks daily, 24.4% had fast food three or more days per week, and 12.7% had sustained multiple injuries in the past year. One in seven students (14.7%) reported WISD, 17.2% among females and 9.6% among males (see Table 1).

### 4.2. Associations with Worry-Induced Sleep Disturbance

In adjusted multivariable logistic regression analysis, 16 years and older (Adjusted Odds Ratio=AOR: 1.48, 95% Confidence Interval=CI: 1.11–1.96), loneliness (AOR: 4.99, 95% CI: 3.82–6.53), having no close friends (AOR: 2.18, 95% CI: 1.35–3.53), infrequent bullying victimization (AOR: 1.56, 95% CI: 1.13–2.16), parents disrespected privacy (AOR: 1.57, 95% CI: 1.06–2.33), current tobacco use (AOR: 1.39, 95% CI: 1.03–1.89), ever cannabis use (AOR: 2.63, 95% CI: 1.24–5.59), high leisure-time sedentary behaviour (AOR: 2.02, 95% CI: 1.43–2.84) and having sustained multiple serious injuries (past year) (AOR: 2.20, 95% CI: 1.53–3.17) were positively and male sex (AOR: 0.44, 95% CI: 0.33–0.60) was negatively associated with WISD. In addition, in unadjusted univariate analysis, mostly or always feeling hungry (or low economic status), school truancy, having been physically attacked, frequently in physical fight (past year), low peer support, parental emotional neglect, parents never check homework, ever drunk and frequent soft drink intake were positively, and infrequent fast food intake was negatively, associated with WISD (see Table 2).

## 5. Discussion

The investigation aimed to estimate the prevalence and correlates of WISD in school adolescents in Lebanon. The prevalence of past 12-month WISD (14.7%) in this study, which was higher than in the 2011 Lebanon GSHS (10.3%) and globally (7.8%), but was lower than in other countries in the Middle East region, e.g., Kuwait (18.9%) and Qatar (17.9%) [8]. Previous studies among adolescents in the Lebanon confirm the high prevalence of sleep disturbance and anxiety disorders [9,10], calling for school sleep and mental health programmes in this adolescent population in Lebanon.

The study showed that being female increased the likelihood of WISD, which was also found in some previous investigations (e.g., [19]). Generally, girls may be more vulnerable to WISD than boys because of different coping styles in response to stressors [13,24]. The study showed that older age increased the likelihood of WISD. Similar results were found in a study among adolescents in Tanzania [14]. Compared to younger adolescents, in older adolescents worry becomes more central anticipating negative outcomes [6], leading to sleep disturbance. In addition, older adolescents in this study engaged more likely in tobacco use, cannabis use, ever drunk, truancy, sedentary behaviour, passive smoking, were more lonely, had no close friends and had multiple injuries than younger adolescents (analysis not shown), which may have contributed to increased WISD among older adolescents [5].

Consistent with former research [12], this survey showed that loneliness and having no close friends were associated with WISD. In a review, Chorney et al. [25] found that there is “a significant symptom overlap between anxiety, depression, and sleep” among children and adolescents. School programmes may want to involve programmes, such as social skills and social support building, to reduce loneliness among adolescents in order to reduce WISD [12]. In line with former research findings [12,13], this survey showed that exposure to interpersonal violence (bullying victimization and, in unadjusted analysis, being physically attacked and frequently involved in physical fighting) increased the odds of WISD. Students exposed to interpersonal violence victimization may worry about further or future victimization, impacting negatively on their sleep [12,26].

Several social-environmental factors (parental disrespect of privacy and in unadjusted analysis passive smoking, low peer support, low parental support, experience of hunger and school truancy) were found to be associated with WISD. These results are consistent with various previous investigations [13,14,15,16] and call for programmes improving parental and peer support.

In terms of health risk behaviours, high sedentary behaviour, having experienced multiple serious injuries, current tobacco use and ever cannabis use increased the likelihood of having WISD. These findings concur with previous studies [8,14,18,19]. In a systematic review, Yang et al. [27] showed that prolonged sedentary behaviour was associated with sleep disturbance. Since this study did not assess the type of sedentary behaviour, for example, screen time, we are not able to show the potentially negative effects of screen time on sleep outcomes [28]. One of the mechanisms by which sedentary behaviour may increase WISD is via inflammatory processes [8]. For example, in a randomized controlled intervention, “a one-week sedentary behaviour-inducing intervention had deleterious effects on anxiety in an active, young adult population” [29]. Possible explanations for the association between multiple injuries and WISD have been described by Lam and Yang [30] in terms of an accumulation of sleep debt via inadequate sleep over time. Sleep debt may lead to sleepiness, “effects a physiologic response that mimics the state of depression” and reduces cognitive performance, all of which increase vulnerability to multiple injuries [30,31]. Unlike some former investigations (e.g., [20]), this survey did not show a significant association between soft drink consumption, fast food intake and WISD.

## 6. Study Limitations

The study limitations include this investigation‘s cross-sectional design, the inclusion of only school adolescents and the self-reporting of data. An additional limitation was that the GSHS in Lebanon only assessed WISD with one item and did not assess help-seeking behaviours for WISD.

## 7. Conclusions

The study found among school-going nationally representative adolescents in Lebanon that one in seven students reported WISD. Several risk factors, including older age, female sex, psychosocial distress (loneliness, having no close friends, and infrequent bullying victimization), social-environmental factors (parents disrespected privacy), and health risk behaviour (current tobacco use, ever cannabis use, high leisure-time sedentary behaviour and having sustained multiple serious injuries), were identified for WISD. Multi-level interventions are needed targeting psychosocial distress, social-environmental factors and health risk behaviours to prevent WISD in this adolescent school population.

## 8. Ethics Approval and Consent to Participate

The present study was based on an analysis of the Lebanon 2017 GSHS survey dataset freely available online with all identifier information detached. The Lebanon 2017 GSHS was approved by the Lebanon Ministry of Education and Higher Education and the World Health Organization. Therefore, the permission and ethical approval for the present analysis was automatically deemed unnecessary. Moreover, during the GSHS survey, written assent attached to a questionnaire was obtained from all eligible participants before filling the questionnaire.

## Figures and Tables

**Table 1 behavsci-10-00148-t001:** Sample and worry-induced sleep disturbance characteristics among adolescents in Lebanon.

Variable	Sample	Worry Induced Sleep Disturbance
	N (%)	N (%)
**Socio-Demographics**		
**All**	**5708 (100)**	**839 (14.7)**
Age in years		
13 or less	1543 (29.1)	143 (8.8)
14–15	1878 (35.5)	284 (14.5)
16 or more	5692 (35.5)	406 (17.1)
Gender		
Female	3370 (56.9)	605 (17.2)
Male	2330 (43.1)	233 (9.6)
Psychosocial distress		
No close friends	268 (4.6)	86 (34.2)
Loneliness	737 (13.1)	335 (45.3)
Bullied in past month		
0 days	4651 (85.0)	592 (11.4)
1 or 2 days	509 (9.7)	100 (20.7)
3–30 days	301 (5.3)	97 (28.4)
Physically attacked in past year		
0 times	4561 (80.1)	575 (11.7)
1 time	523 (9.1)	108 (18.7)
2 or more times	606 (10.7)	154 (23.1)
In physical fight in past year		
0 times	3684 (65.5)	477 (12.2)
1 time	814 (14.5)	128 (13.2)
2 or more times	1147 (20.1)	224 (17.7)
Social-environmental factors		
Mostly/always feeling hungry	163 (2.6)	43 (26.0)
Low peer support	978 (20.0)	191 (19.0)
Parental emotional neglect	501 (11.1)	103 (17.2)
Parents never check home work	1337 (23.9)	238 (16.8)
Parents disrespect privacy	388 (7.6)	87 (23.7)
Passive smoking in past week		
0 days	1571 (28.8)	166 (10.3)
1–6 days	2031 (34.4)	253 (11.1)
All 7 days	2005 (36.8)	401 (19.3)
School truancy (past month)		
0 days	4406 (82.6)	622 (13.0)
1–2 days	610 (11.7)	113 (18.2)
3 or more days	288 (5.6)	63 (20.7)
Health risk behaviours		
Ever drunk	587 (10.9)	109 (17.9)
Current tobacco use	1987 (35.9)	433 (20.4)
Ever cannabis use	141 (2.5)	41 (31.1)
Leisure time sedentary behaviour/day		
<3 h	2979 (59.2)	363 (10.9)
3–4 h	1208 (21.2)	190 (13.2)
5–8 h	744 (12.7)	127 (17.4)
>8 h	394 (7.0)	114 (29.2)
Soft drink intake in a day		
0	2969 (49.2)	421 (13.2)
1	1288 (24.7)	155 (11.4)
2	711 (13.1)	122 (15.5)
3 or more	717 (13.0)	138 (17.9)
Fast food intake in past week		
0 days	1346 (23.9)	209 (15.6)
1 day	1730 (30.3)	230 (11.6)
2 days	1224 (21.4)	157 (11.9)
3 or more days	1377 (24.4)	241 (16.2)
Injury in past 12 months		
0 times	3332 (64.7)	347 (9.9)
1 time	1130 (22.6)	221 (17.1)
2 or more times	654 (12.7)	182 (25.1)

**Table 2 behavsci-10-00148-t002:** Associations with worry-induced sleep disturbance.

Variable	Unadjusted Odds Ratio (95% CI)	Adjusted Odds Ratio (95% CI)
Socio-Demographics		
Age in years		
13 or less	1 (Reference)	1 (Reference)
14–15	1.75 (1.41, 2.17) ***	1.53 (1.14, 2.05) **
16 or more	2.13 (1.73, 2.64) ***	1.48 (1.11, 1.96) **
Gender		
Female	1 (Reference)	1 (Reference)
Male	0.51 (0.41, 0.64) ***	0.44 (0.33, 0.60) ***
Psychosocial distress		
No close friends	3.56 (2.54, 5.00) ***	2.18 (1.35, 3.53) **
Loneliness	7.96 (6.37, 9.95) ***	4.99 (3.82, 6.53) ***
Bullied in past month		
0 days	1 (Reference)	1 (Reference)
1 or 2 days	2.04 (1.58, 2.64) ***	1.56 (1.13, 2.16) **
3–30 days	3.10 (2.19, 4.39) ***	1.69 (0.90, 3.17)
Physically attacked in past year		
0 times	1 (Reference)	1 (Reference)
1 time	1.74 (1.34, 2.26) ***	1.18 (0.80, 1.74)
2 or more times	2.26 (1.94, 2.63) ***	1.53 (0.99, 2.37)
In physical fight in past year		
0 times	1 (Reference)	1 (Reference)
1 time	1.09 (0.90, 1.32)	1.02 (0.76, 1.36)
2 or more times	1.54 (1.23, 1.94) ***	0.95 (0.63, 1.43)
Social-environmental factors		
Mostly/always feeling hungry	2.31 (1.55, 3.45) ***	1.39 (0.83, 2.31)
Low peer support	1.60 (1.25, 2.04) ***	1.06 (0.73, 1.53)
Parental emotional neglect	1.33 (1.00, 1.77) *	1.10 (0.76, 1.60)
Parents never check home work	1.35 (1.10, 1.66) **	1.12 (0.90, 1.40)
Parents disrespect privacy	2.07 (1.54, 2.77) ***	1.57 (1.06, 2.33) *
Passive smoking in past week		
0 days	1 (Reference)	1 (Reference)
1–6 days	1.08 (0.87, 1.35)	0.93 (0.73, 1.19)
All 7 days	2.08 (1.73, 2.50) ***	1.25 (0.91, 1.71)
School truancy (past month)		
0 days	1 (Reference)	1 (Reference)
1–2 days	1.49 (1.20, 1.86) ***	1.24 (0.91, 1.71)
3 or more days	1.75 (1.23, 2.47) **	1.28 (0.83, 1.97)
Health risk behaviours		
Ever drunk	1.45 (1.09, 1.92) *	0.81 (0.47, 1.39)
Current tobacco use	2.26 (1.76, 2.91) ***	1.39 (1.03, 1.89) *
Ever cannabis use	3.00 (2.06, 4.38) ***	2.63 (1.24, 5.59) *
Leisure time sedentary behaviour/day		
<3 h	1 (Reference)	1 (Reference)
3–4 h	1.24 (1.02, 1.52) *	1.23 (0.93, 1.62)
5–8 h	1.71 (1.36, 2.16) ***	1.21 (0.88, 1.65)
>8 h	3.36 (2.65, 4.26) ***	2.02 (1.43, 2.84) ***
Soft drink intake in a day		
0	1 (Reference)	1 (Reference)
1	0.84 (0.70, 1.02)	0.84 (0.66, 1.07)
2	1.21 (0.91, 1.61)	1.03 (0.71, 1.48)
3 or more	1.43 (1.10, 1.87) **	1.02 (0.63, 1.65)
Fast food intake in past week		
0 days	1 (Reference)	1 (Reference)
1 day	0.71 (0.55, 0.93) *	0.86 (0.63, 1.16)
2 days	0.73 (0.58, 0.93) *	0.71 (0.47, 1.08)
3 or more days	1.05 (0.78, 1.41)	0.90 (0.61, 1.31)
Injury in past 12 months		
0 times	1 (Reference)	1 (Reference)
1 time	1.88 (1.54, 2.30) ***	1.87 (1.35, 2.58) ***
2 or more times	3.06 (2.49, 3.76) ***	2.20 (1.53, 3.17) ***

*** *p* < 0.001; ** *p* < 0.01; * *p* < 0.05; CI = Confidence Interval.

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
