# Peer review of "Prevalence of Worry-Induced Sleep Disturbance and Associated Factors among a National Sample of In-School Adolescents in Lebanon"

_behavsci, 2020, doi:10.3390/bs10100148_

Round 1
Reviewer 1 Report
The authors are trying to publish an article where they studied the prevalence and factors associated with stress-induced sleep disorder among adolescents.
The manuscript has serious flaws that need to be addressed.
1.)In the background, I am not able to get a clear understanding of how they define worry induced sleep disturbance. When I went through the reference, they cited (no 7), I could not find any definition of worry induced sleep disturbance. I am not able to get a sense of what they mean by worry induced sleep disturbance. For example, inline 42, they mention that worry is a cognitive component of anxiety, but later inline 44, they said WISD includes “symptoms 44 of depression, anxiety, stress, and somatic complaints”. I would recommend to clearly define what they are looking for in this population and explain it clearly. I would also recommend choosing medical terminologies like stress for worry and disorder for the disturbance.
2.)In the methods section, how do they diagnose WISD on this patient population? what criteria were used?
2.)When I went through the questionnaire, they did not mention the sleep quality and duration between these 2 groups. I am not sure how they labeled the other group as WISD patients and how they assessed ?.
3.) In the results section, they did not mention whether it was a univariate analysis or multivariate analysis. No p values were mentioned.
Author Response
Reviewer I:
The authors are trying to publish an article where they studied the prevalence and factors associated with stress-induced sleep disorder among adolescents.
The manuscript has serious flaws that need to be addressed.
1.)In the background, I am not able to get a clear understanding of how they define worry induced sleep disturbance. When I went through the reference, they cited (no 7), I could not find any definition of worry induced sleep disturbance. I am not able to get a sense of what they mean by worry induced sleep disturbance. For example, inline 42, they mention that worry is a cognitive component of anxiety, but later inline 44, they said WISD includes “symptoms 44 of depression, anxiety, stress, and somatic complaints”. I would recommend to clearly define what they are looking for in this population and explain it clearly. I would also recommend choosing medical terminologies like stress for worry and disorder for the disturbance.
Response: This is corrected, as in below
Worry-induced sleep disturbance (WISD) is defined as mostly or always “worried about something that you could not sleep at night in the past 12 months” [7]. Among children and adolescents, in moderation, worry may be adaptive, but if worries persist over time they can cause clinical concern, and pathological worry has been associated with various problems, including poor academic performance and disrupted sleep [8].
2.)In the methods section, how do they diagnose WISD on this patient population? what criteria were used?
2.)When I went through the questionnaire, they did not mention the sleep quality and duration between these 2 groups. I am not sure how they labeled the other group as WISD patients and how they assessed ?.
Response to above, as below:
Worry-induced sleep disturbance was assessed with the question, “During the past 12 months, how often have you been so worried about something that you could not sleep at night?” Response options were never, rarely, sometimes, most of the time or always. Most or the time or always was defined as worry-induced sleep disturbance, while never, rarely or sometimes was defined as no worry-induced sleep disturbance, in line with previous research [7].
3.) In the results section, they did not mention whether it was a univariate analysis or multivariate analysis. No p values were mentioned
Response: This added, as below
In adjusted multivariable logistic regression analysis, 16 years and older (AOR: 1.48, 95% CI: 1.11-1.96), loneliness (AOR: 4.99, 95% CI: 3.82-6.53), having no close friends (AOR: 2.18, 95% CI: 1.35-3.53), infrequent bullying victimization (AOR: 1.56, 95% CI: 1.13-2.16), parents disrespected privacy (AOR: 1.57, 95% CI: 1.06-2.33), current tobacco use (AOR: 1.39, 95% CI: 1.03-1.89), ever cannabis use (AOR: 2.63, 95% CI: 1.24-5.59), high leisure-time sedentary behaviour (AOR: 2.02, 95% CI: 1.43-2.84) and having sustained multiple serious injuries (past year) (AOR: 2.20, 95% CI: 1.53-3.17) were positively and male sex (AOR: 0.44, 95% CI: 0.33-0.60) was negatively associated with WISD. In addition, in unadjusted univariate analysis, mostly or always feeling hungry (or low economic status), school truancy, having been physically
Reviewer 2 Report
In this manuscript, the authors reported a study that aimed to estimate the prevalence and correlates of worry induced sleep disturbance (WISD) among adolescent school children in Lebanon. Although the matter of this manuscript is of potential interest and carried out in a large sample of individuals, I think that this study is very basic in its design and probably not so strong from statistical point of view to state such a conclusion. Therefore, I should strongly encourage the authors to revise the manuscript and correct mistakes. Here below I reported the points that must be addressed:
- ABSTRACT: I should suggest the authors to define the state of art and to focus on the aim of the study. Also the conclusions should be articulated in a better manner.
- RESULTS: The results of the study are not well-presented, it’s a list of names without any organization. Data reported in the text did not correspond to those in the tables. Moreover, the sums and percentages did not add up (e.g.: females 3370 and males 2330 = 5700 and not 5708). The tables are confusing, so please reorganize them in a better way.
- What does it mean the sentence “more than one”? It’s not clear.
- Table 2: please put “-“ instead of “,” in the CI values, because the interval of confidence is a range.
- CONCLUSIONS: I should suggest to better discuss the great implications of the study in this section.
Author Response
In this manuscript, the authors reported a study that aimed to estimate the prevalence and correlates of worry induced sleep disturbance (WISD) among adolescent school children in Lebanon. Although the matter of this manuscript is of potential interest and carried out in a large sample of individuals, I think that this study is very basic in its design and probably not so strong from statistical point of view to state such a conclusion. Therefore, I should strongly encourage the authors to revise the manuscript and correct mistakes. Here below I reported the points that must be addressed:
- ABSTRACT: I should suggest the authors to define the state of art and to focus on the aim of the study. Also the conclusions should be articulated in a better manner.
Response: Both are added
- RESULTS: The results of the study are not well-presented, it’s a list of names without any organization. Data reported in the text did not correspond to those in the tables.
Response: Results are reorganized, as below
In terms of psychosocial distress, 13.1% of adolescents were lonely, 4.6% had no close friends, 5.3% had frequently been bullied (≥3 days/past month), 10.7% had frequently been attacked (≥2 times/past year) and 20.1% had frequently been in a physical fight (≥2 times/past year). In relation to social-environmental factors, 2.6% of students were mostly or always hungry, 36.8% were exposed to daily secondary smoke, 17.3% were school truant, 20.0% had low peer support, 11.1% experienced parental emotional neglect, 23.9% had parents who never checked on their home work, and 7.6% had parents who mostly or always disrespected their privacy. Regarding health risk behaviours, 35.9% of the students used currently tobacco, 10.9% had ever been drunk, 2.5% had ever used cannabis, 13.0% had daily three or more soft drinks, 24.4% had weekly on three or more days fast food, and 12.7% had multiple injuries in the past year
Moreover, the sums and percentages did not add up (e.g.: females 3370 and males 2330 = 5700 and not 5708).
Response: The percentages do add up, while the actual number may not add up because of missing cases.
The tables are confusing, so please reorganize them in a better way.
Response: The description of the results is reorganized, as above
- What does it mean the sentence “more than one”? It’s not clear.
Response: For example in below, it clearly says “more than one in three students” means 35.9%
More than one in three students (35.9%)
- Table 2: please put “-“ instead of “,” in the CI values, because the interval of confidence is a range.
Response: This could be done, but according to below it is “,” in the Table, but in the text “-“, as done, accordingly
The APA 6 style manual states (p. 117): “ When reporting confidence intervals, use the format 95% CI [LL, UL] where LL is the lower limit of the confidence interval and UL is the upper limit. ” For example, one might report: 95% CI [5.62, 8.31].
- CONCLUSIONS: I should suggest to better discuss the great implications of the study in this section.
Response: added, as below
Conclusion
The study found among school-going nationally representative adolescents in Lebanon that one in seven students reported WISD. Several risk factors, including older age, female sex, psychosocial distress (loneliness, having no close friends, and infrequent bullying victimization), social-environmental factors (parents disrespected privacy), and health risk behaviour (current tobacco use, ever cannabis use, high leisure-time sedentary behaviour and having sustained multiple serious injuries), were identified for WISD. Multi-level interventions targeting psychosocial distress, social-environmental factors and health risk behaviours to prevent AISD in this adolescent school population.
Round 2
Reviewer 1 Report
The authors have made the recommended changes.
I would recommend English editing for the better flow of the content.
Reviewer 2 Report
The authors significantly improved their manuscript that is now suitable for publication.